



# Aerosol reductions outweigh circulation changes for future improvements in Beijing haze

Liang Guo[1], Laura J. Wilcox[1,2], Massimo Bollasina[3], Steven T. Turnock[4], Marianne T. Lund[5], and Lixia Zhang[6]

[1]National Centre for Atmospheric Science, United Kingdom
[2]Department of Meteorology, University of Reading, Reading, United Kingdom
[3]School of Geosciences, Grant Institute, University of Edinburgh, Edinburgh, United Kingdom
[4]Met Office Hadley Centre, Exeter, Uinted Kingdom
[5]Center for International Climate and Environmental Research, Oslo, Norway
[6]State Key Laboratory of Numerical Modeling for Atmospheric Sciences and Geophysical Fluid Dynamics, Institute of Atmospheric Physics, Chinese Academy of Sciences, Beijing, China

**Correspondence:** Liang Guo (l.guo@reading.ac.uk)

**Abstract.** Despite local emission reductions, severe haze events remain a serious issue in Beijing. Previous studies have suggested that both greenhouse gas increases and aerosol decreases are likely to increase the frequency of weather patterns conducive to haze events. However, the combined effect of atmospheric circulation changes and aerosol and precursor emission changes on Beijing haze remains unclear. We use the Shared Socioeconomic Pathways (SSPs) to explore the effects of aerosol and greenhouse gas emission changes on both haze weather and Beijing haze itself. We confirm that the occurrence of haze weather patterns is likely to increase in future under all SSPs, and show that even though aerosol reductions play a small role, greenhouse gas increases are the main driver, especially during the second half of the 21st century. However, the severity of the haze events decreases on decadal timescales by as much as 70% by 2100. The main influence on the haze itself is the reductions in local aerosol emissions, which outweigh the effects of changes in atmospheric circulation patterns. This demonstrates that aerosol reductions are beneficial, despite their influence on the circulation.

## 1 Introduction

Over a million premature deaths in China were attributed to poor air quality in 2010, accounting for over 30% of the mortality due to air pollution worldwide (Lelieveld et al., 2015; Zhang et al., 2017). Ambitious clean air policies, designed to address this serious issue, have resulted in dramatic reductions in Chinese emissions of the particulate matter and gases that contribute to poor air quality since 2008 (Zheng et al., 2018; Li et al., 2017). Sulphur dioxide ($SO_2$) and black carbon (BC) emissions declined by 70% and 35% respectively between 2008 and 2017. However, haze, which is defined as the occurrence of low visibility ($\leq$ 10km) due to accumulation of fine particulate matter in the air (Wu, 2011), remains severe in China (Xue et al., 2019; Le et al., 2020; Chen et al., 2013; Rohde and Muller, 2015).

There are potentially a number of factors at play that may cause haze to remain a frequent occurrence despite the decrease in aerosol and precursor emissions. Meteorological conditions (Zhang et al., 2020b) play an important role in modulating the



occurrence and persistence of haze events. In fact, atmospheric circulation patterns conducive to haze events may increase in the future as the climate warms (Cai et al., 2017; Pei and Yan, 2018). Continued haze events could also be due to a lack of mitigation of secondary aerosols (Huang et al., 2014; Zheng et al., 2018; An et al., 2019; Le et al., 2020). In 2020, severe haze was formed during the COVID-19 lockdown despite emission reductions of up to 90% (Le et al., 2020).

A metric that has been recently introduced to characterise the meteorological conditions associated with haze over north China, and Beijing in particular, is the haze weather index (HWI) (Cai et al., 2017). The HWI accounts for the role of circulation and vertical stratification, the two main meteorological factors involved in haze occurrence. By analysing 15 climate models under the Representative Concentration Pathway 8.5 (RCP8.5 , frequently referred to as the "business as usual" scenario and featuring a large future increase in greenhouse gas emissions), Cai et al. (2017) predicted a 50% increase in the frequency and

a 80% increase in the persistence of these meteorological conditions in the second half of the 21st century compared to the 20th century as a consequence of warming and circulation changes induced by greenhouse gases.

    Aerosol reductions may also modulate meteorological conditions and, subsequently, lead to increases in HWI (Jiang et al., 2017; Liu et al., 2019; Zhang et al., 2020a). Affected by aerosols, and also affecting their accumulation in northeast China, the HWI is intrinsically intertwined with aerosol emissions and haze formation. Yet, meteorological conditions represent a

key uncertainty in future climate projections and a large source of discrepancy among models (Callahan and Mankin, 2020). New future scenarios used in the Sixth Coupled Model Intercomparison Project (CMIP6), the Shared Socioeconomic Pathways (SSPs), are designed to cover a wide range of narratives of socioeconomic development and energy consumption in the 21st century, and capture a much wider range of uncertainty in aerosol emission pathways (Figure 1) than the Representative Concentration Pathways used in CMIP5 (Riahi et al., 2017). This presents an opportunity to explore the relative roles of

aerosol and greenhouse gases in driving changes in the HWI, and the interplay between the effects of aerosol changes on the atmospheric circulation and on the composition of the haze itself. In this study we examine whether, in scenarios with rapid aerosol reductions, increases in haze frequency or severity may not be as large as trends in circulation-based metrics alone suggest.

## 2   Data and Methods

### 2.1   Future emission pathways

Future aerosol and greenhouse gas pathways in CMIP6 are described by the Shared Socioeconomic Pathways (SSPs), which represent a range of socioeconomic narratives (Riahi et al., 2017), designated by the first digit of the three numbers in the scenario names: 1 describes a sustainable development; 2 is a medium change narrative; 3 describes a regional rivalry situation; and 5 describes a fossil-fuel development. The following two digits separated by a decimal point show the global mean radiative

forcing by 2100 in $W/m^2$. The air pollution pathways have been designed to be consistent with the SSPs, but also make specific assumptions about the stringency of air quality policy. We use four SSPs in this work: 1-2.6, 2-4.5, 3-7.0 and 5-8.5. Figure 1 shows the time series of area-averaged $SO_2$, BC and $CO_2$ emissions from East Asia, South Asia, and the globe in four SSPs, during 2015-2100. Over East Asia, both $SO_2$ and black carbon (BC) emissions will decrease from present-day to the end



of the century in most SSPs, except in SSP 3-7.0, wherein emissions continue to increase until 2040. The change in aerosol emissions over East Asia shows a rapid reduction in SSP 1-2.6, and comparable moderate reductions in SSPs 2-4.5 and 5-8.5. In SSP 1-2.6, aerosol emissions reduce rapidly until 2050, and then flatten until 2100. In SSPs 2-4.5 and 5-8.5, emissions reduce gradually and reach the minima by 2100. Radiative forcing at the end of the 21st century varies consistently with the amount of global $CO_2$ emissions.

## 2.2 Air quality and circulation indices

The Haze Weather Index (HWI) is defined following Cai et al. (2017), using a combination of meteorological variables that capture the key characteristics of weather patterns conducive to severe haze in and around Beijing.

$$HWI = \Delta T_{\mathrm{p}} + V_{850} + \Delta U_{500}, \tag{1}$$

where $\Delta T$ is the temperature difference between 850 hPa (averaged over $32.5° - 45°$N, $112.5° - 132.5°$E) and 250 hPa (averaged over $37.5° - 45°$N, $122.5° - 137.5°$E); $V_{850}$ is the 850-hPa meridional wind averaged over ($30° - 47.5°$N, $115° - 130°$E); $\Delta U$ is the latitudinal difference of the 500 hPa zonal wind between areas north ($42.5° - 52.5°$N, $110° - 137.5°$E) and south ($27.5° - 37.5°$N, $110° - 137.5°$E) of Beijing. These three terms represent, respectively, the vertical stratification of the atmospheric column over Beijing, the circulation anomaly in the region, and the magnitude of the near-surface wind. The regions used in the calculation are shown in Figure 2. For each of the three terms, anomalies are calculated relative to the mean over the considered time periods. The time series is then standardised by the standard deviation over the same time period. The HWI is then calculated as the standardised anomaly of the sum of the three terms. $HWI > 0$ designates a meteorological pattern conducive to a haze event.

The strength of the East Asian Winter Monsoon is measured by the WCI index (Wang and Chen, 2014), which considers the sea level pressure (SLP) differences between Siberia and the northern Pacific, as well as between Siberia and the Maritime Continent:

$$WCI = (SLP_{sib} - SLP_{np}) + (SLP_{sib} - SLP_{mc}), \tag{2}$$

where $SLP_{xx}$ is the standardised SLP anomaly over Siberia (sib; $40° - 60°$N, $70° - 120°$E), the northern Pacific (np; $30° - 50°$N, $140° - 170°$E) or the Maritime Continent (mc; $20°$S $- 10°$N, $110° - 160°$E). The first term represents a zonal pressure difference, while the second represents a meridional difference. WCI is the standardised anomaly of the sum of these differences. $WCI > 0$ indicates a strong winter monsoon, which makes Beijing haze events less likely.

Both indices quantify the strength of the East Asian Winter Monsoon, although HWI is designed to capture specific patterns over Beijing. By using two indices based on different variables, any results will be considered to be more robust if they are consistent across the indices, as the different variables involved should be differently affected by model biases. HWI is negatively related to WCI, as a positive HWI indicates a weak winter monsoon (i.e., a negative WCI). To make the comparison straightforward, the sign of WCI will be reversed and denoted as WCI* throughout, so that positive values of both indices indicate an increased likelihood of haze.



### 2.3 Data

Monthly pressure level variables are used to calculate seasonal-mean HWI and WCI* during December-January-February (DJF). In its original definition (Cai et al., 2017), HWI was computed from daily data. Yet, despite a slightly reduced magnitude, values of the HWI using monthly data are consistent with those based on daily data (Zhang et al., 2020a). Since the use of monthly data allows us to use a greater number of CMIP6 models, monthly HWI is computed and analysed here. Any model that has data available in both the CMIP6 historical experiment and one of the SSPs is included. As a result, 17 models are used (Table 1). Model ensemble sizes in CMIP6 range from 1 member per model to 50 members. In the analysis of future anomalies in HWI and WCI*, the multi-model mean is calculated from individual model means to give an equal weight to each model. However, when evaluating model variability, individual ensemble members are used. For model evaluation, and the reference period for the calculation of future anomalies, the present-day is defined as the period 1979-2014.

In addition to the analysis of circulation indices, we evaluate the severity of haze events based on variables related to aerosol concentrations in the wider Beijing region ($37.5° − 42.5°$N, $114° − 119°$E), as such variables give a more direct link to the haze itself: aerosol optical depth (AOD) at 550 nm, and the concentration of surface particulate matter less than 2.5 $\mu$m in diameter ($PM_{2.5}$). $PM_{2.5}$ concentrations are calculated following Turnock et al. (2020) as:

$$PM_{2.5} = BC + OA + SO_4 + (0.25 \times SS) + (0.1 \times DU), \tag{3}$$

where $PM_{2.5}$ is the sum of the dry aerosol mass mixing ratio of $BC$, total organic carbon ($OA$ - both primary and secondary sources), sulphate ($SO_4$), sea salt ($SS$) and dust ($DU$) from the lowest model level. A scaling factor of 0.25 for $SS$ and 0.1 for $DU$ has been used to calculate the approximate contribution from these components to the fine size fraction ($< 2.5\mu$m), and is applied to data from all models. Unfortunately, the availability of aerosol variables from CMIP6 models is limited compared to the atmospheric variables. Models and experiments that include AOD and/or $PM_{2.5}$ are highlighted in Table 1.

The fifth generation European Center for Medium-Range Weather Forecasts atmospheric reanalysis (ERA5, Hersbach et al., 2020) is used to evaluate the present-day distribution of HWI and WCI*, and the constituent variables, in CMIP6 models. Compared to previous reanalysis versions, ERA5 has improvements in model physics, core dynamics and data assimilation.

## 3 Present-day East Asian Winter climate in CMIP6

The CMIP6 multi-model mean captures the pattern and magnitude of the key features of the East Asian Winter climate reasonably well (Figure2). Compared to ERA5, the CMIP6 multi-model mean shows a cooler lower-troposphere (Figure 2c), a southward shifted mid-latitude westerly jet (Figure 2f), and a stronger low-level prevailing northerly (Figure 2i), all of which are indicative of a stronger East Asian Winter Monsoon.

The reanalysed (ERA5) and simulated (CMIP6) present-day (1979-2014) distribution of the magnitude of HWI and WCI* during DJF are compared in Figure 3. CMIP6 models are consistent with ERA5, in terms of both mean and spread. A Kolmogorov-Smirnov test is applied, which indicates that indices from ERA5 and CMIP6 are drawn from the same distribution. A closer inspection of Figure 3 shows that the CMIP6 models tend to simulate a stronger winter circulation than in





ERA5, as evident by the slight shift to the left of distributions with respect to the reanalysis, which is consistent with the large-scale comparison shown in Figure 2. However, neither difference is significant. The skillful representations of present-day observations of HWI and WCI* in CMIP6 justifies the use of these models to estimate future changes in Beijing haze.

## 4   Haze changes throughout the 21st century

Future HWI and WCI* changes are calculated as deviations from the present-day climatological values and presented as 10-year averages for 2025-2034, 2035-2044, 2045-2054 and 2090-2099 (Figure 4). These periods are chosen to capture the near-future when there is a rapid decline in aerosol emissions in SSP 1-2.6 and large differences in aerosol pathways across the SSPs, and the end of the century when differences between the SSPs are dominated by $CO_2$ emissions.

In all future periods, HWI is larger than in the present day (Figure 4a), indicating that weather patterns conducive to haze will occur more frequently, be more severe, or both. WCI* is also greater, indicating a weaker winter monsoon circulation, and more favourable conditions for haze.

The positive HWI anomaly is larger in SSPs 3-7.0 and 5-8.5 relative to other SSPs by 2045-2054 and reaches its maximum by 2090-2099 as global $CO_2$ emissions continue to increase in both scenarios. Similar changes occur in WCI* by 2090-2099 (Figure 4b). Furthermore, the order of the medians of the both HWI and WCI* anomalies across the different scenarios during 2090-2099 correspond to global accumulative $CO_2$ emissions (Figure 1), with the anomalies for SSPs 3-7.0 and 5-8.5 larger than in other SSPs. This indicates the dominance of $CO_2$ in changing HWI and the winter monsoon during the second half of the 21st century. The thermal component ($\Delta T_{\mathrm{vertical}}$ in Eq. 1) of HWI makes the largest contribution to the future increases in HWI (Figure A1b), and shares these characteristics of $CO_2$-dominated changes.

In the first half of the 21st century, the competition between the response to aerosol and $CO_2$ changes are visible. During 2025-2034, WCI* has the smallest increase in SSP 3-7.0, where aerosol emissions continue to increase. In contrast, WCI* has the largest increase in SSP 1-2.6, where aerosol emissions decrease sharply during the same period. This is consistent with aerosol reductions driving increases in haze indices identified by Zhang et al. (2020a). Although changes in $CO_2$ emissions are different between SSPs 2-4.5 and 5-8.5, reductions in aerosol emissions are similar in this period (2025-2034). This similarity is reflected in the median WCI* anomalies, further suggesting that the aerosol emission pathway influences the relative magnitude of WCI* anomalies during the early 21st century. However, compared to responses to $CO_2$ in the second half of the 21st century, these differences in HWI and WCI* across the SSPs are small and insignificant (Figure A3). HWI changes are similar to but weaker than WCI*, which can be attributed to the larger internal variability in HWI due to the smaller spatial scales of the component terms.

Both HWI and WCI* show that patterns conducive to haze events become more likely with increases in $CO_2$ emissions in the long term. However, without sources of aerosol, the formation of haze is unlikely. The actual change in haze itself depends on changes in both the atmospheric circulation and aerosol concentrations. AOD at 550nm and $PM_{2.5}$ are used as indicators of haze severity. Anomalies relative to the present day for $HWI > 1$ (the threshold for 'haze days' in the present climate) are shown in Figure 5. Changes in future AOD and $PM_{2.5}$ over Beijing correspond to those in aerosol precursor emissions over





East Asia (Figure 1). The mean AOD and PM$_{2.5}$ rapidly reduce to values below the present-day level (by more than 25%) in all scenarios apart from SSP 3-7.0, where the mean AOD and PM$_{2.5}$ are greater than the present-day level until 2050 (up to 35%), consistent with continued increases in local emissions. As indicated by changes in AOD and PM$_{2.5}$, haze events in all scenarios except SSP 3-7.0 become less severe (Figure 5 a and c), despite the concomitant increases in HWI and WCI*. Future changes

in AOD and PM$_{2.5}$ follow the reductions in future aerosol emissions, rather than the increase seen in the circulation metrics, indicating that the reduction in aerosol emissions outweighs the increase in haze weather patterns and dominates changes of future haze events.

HWI, however, still can be useful in predicting future haze events. Figure 5 b and d show the percentage changes in AOD and PM$_{2.5}$ between haze ($HWI > 1$) and contemporary non-haze days ($HWI < 0$). In all periods, AOD and PM$_{2.5}$ have similar

relative anomalies. Except SSP 1-2.6, the anomalies range between $20 \sim 40\%$. This indicates that by using a certain value of HWI to define haze events (for example, $HWI = 1$), air pollutant increases by a certain amount regardless of the value of the baseline. However, as future aerosol emissions will reduce, a larger HWI threshold is likely to be needed to identify circulation patterns likely to cause PM$_{2.5}$ concentrations to exceed dangerous levels.

## 5   Conclusion and Discussion

This study investigated 21st century changes in Beijing haze events using CMIP6 models. Circulation patterns conducive to the formation of haze increase in all future scenarios due to the weakening of East Asian winter monsoon, with a clear relationship with increases in CO$_2$ at the end of the century. Scenarios with the largest CO$_2$ emission have significantly larger increases in two haze weather indices, HWI and WCI*, by 2100. The opposing impacts of aerosols on these patterns can be seen in the near future (2025-2034) in SSP 3-7.0 where aerosol emissions continue to increase, moderating increases in the haze indices.

However, although near-future changes in HWI and WCI* are consistent with differences in aerosol emission across the SSPs, the differences in aerosol pathways are not large enough to result in significant differences between the SSPs.

Future changes in the severity of haze events themselves were evaluated using anomalies in AOD at 550nm and surface PM$_{2.5}$. Despite increases in HWI and WCI*, the severity of Beijing haze decreases with reductions in aerosol and precursor emissions. This shows that the decrease in aerosol emissions under strong air pollution mitigation scenarios outweighs the

continued increase in haze weather patterns and weakening of the winter monsoon. The above findings indicate that using meteorological indices alone to investigate future changes in haze can be misleading, and should be complemented by the analysis of changes in air quality metrics.

We show that reducing aerosol emissions is beneficial for Beijing air quality in the long term, despite their reductions making the atmospheric circulation patterns associated with haze more likely. The severity of haze events reduces most in SSP 1-2.6,

which has the fastest and largest emission reduction, while it reduces the least in SSP 3-7.0, which has the slowest and smallest reduction.



*Acknowledgements.* This work and its contributors Liang Guo, Laura Wilcox, and Massimo Bollasina were supported by the UK-China Research and Innovation Partnership Fund through the Met Office Climate Science for Service Partnership (CSSP) China as part of the Newton Fund.





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





**Figure 1.** Time series of total emissions for BC, SO2 (Tg) and CO2 (Pg) in SSPs1-2.5, 2-4.5, 3-7.0 and 5-8.5 from East Asia, South Asia and the globe. Periods used in analyses are highlighted in the first row panels.





**Figure 2.** Biases in present-day East Asian Winter Monsoon (DJF 1979-2014). (a-c) Zonal mean temperature averaged between $112.5° - 137.5°$E for CMIP6, ERA5 and their difference. (d-f) 500hPa zonal wind. (g-i) 850hPa meridional wind. (j-l) Sea level pressure. Difference that is above 90% confidence level is stippled. Boxes indicate areas from which averaged values are used to compute HWI and WCI.





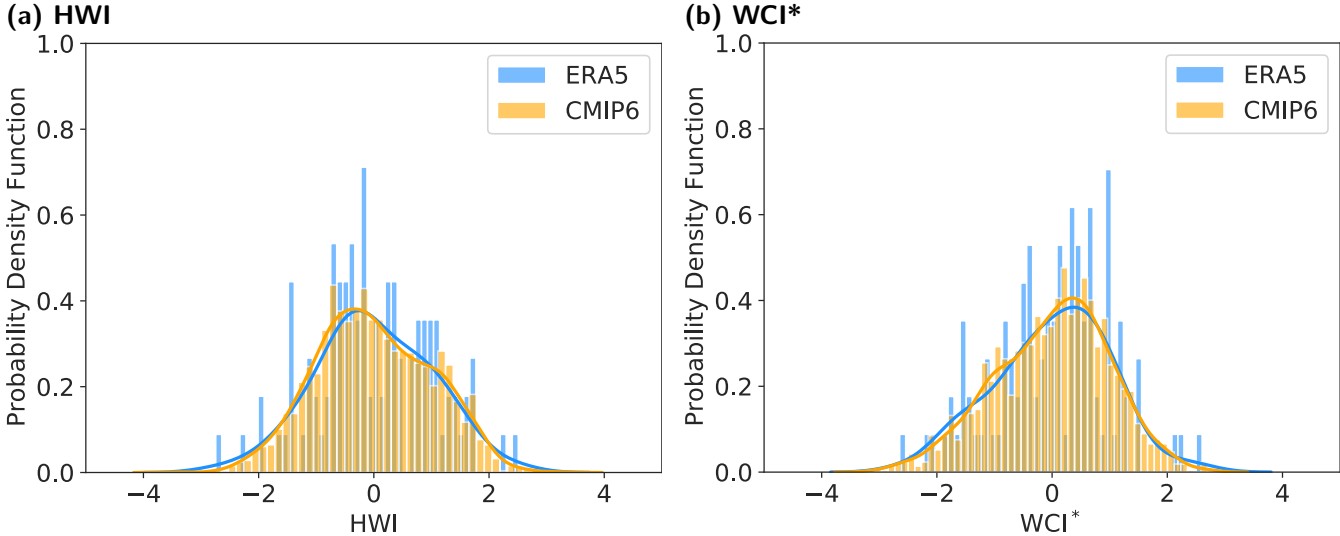

**Figure 3.** Probability distribution of (a) HWI and (b) WCI* calculated from CMIP6 (orange) and ERA5 (azure) for present-day East Asia winter (DJF 1979-2014).



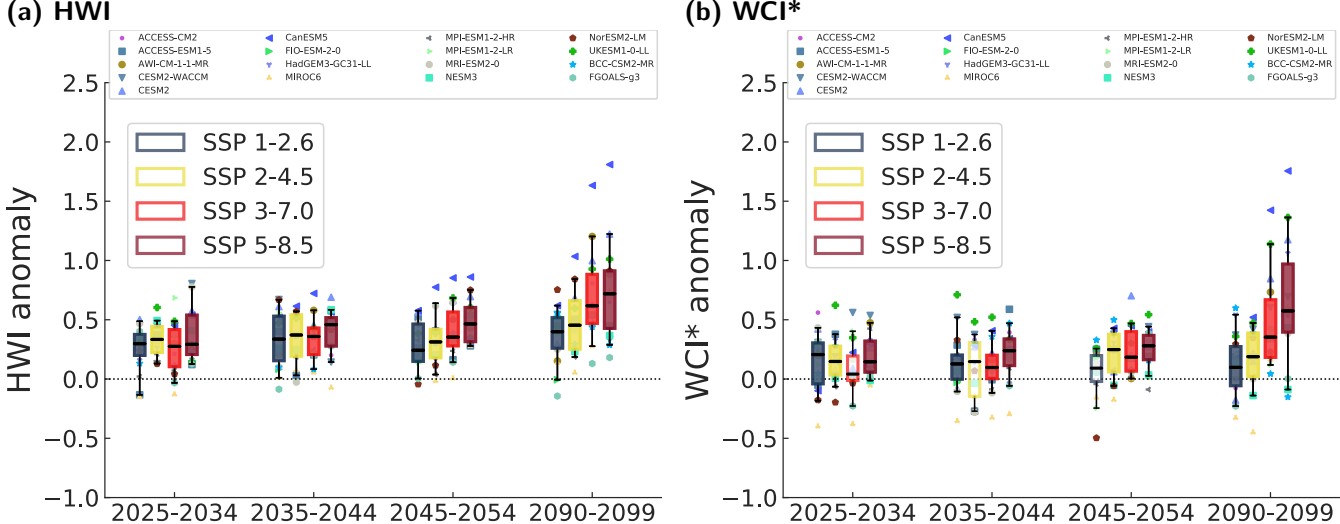

**Figure 4.** Future HWI (a) and WCI* (b) changes from present day (1979-2014). Colour boxes represent the range of between the 1st and the 3rd quartiles of the probability distribution of index anomalies in scenarios (SSPs1-2.6, 2-4.5, 3-7.0 and 5-8.5) during periods of 2025-2034, 2035-2044, 2045-2054 and 2090-2099. Boxes are filled if the change is significant at the 90% confidence level according to a two-tailed t-test. Black bars represent the median of distributions. Black whiskers represent the range of between the 5th and the 95th of the probability distribution of index anomalies. Model mean values are represented by markers with different shape and colour. $WCI^* = -1 \times WCI$. Details in Section Data and Methods.



**Figure 5.** (a) AOD at 550nm anomalies during haze events ($HWI > 1$) compared to present-day averaged over Beijing ($37.5° − 42.5°$N, $114° − 119°$E). (b) AOD at 550nm anomalies during haze events ($HWI > 1$) compared to contemporary non-haze days ($HWI \leq 0$) over Beijing. (c) PM$_{2.5}$ anomalies ($HWI > 1$) compared to present-day averaged over Beijing. (d) PM$_{2.5}$ anomalies compared to contemporary non-haze day ($HWI \leq 0$) over Beijing. Units: %.





**Table 1.** CMIP6 simulation availability. Number of ensemble member used for model and scenario is shown. $a$ denotes aerosol optical depth is available from at least one ensemble member. $p$ denotes $PM_{2.5}$ is available from at least one ensemble member.

| Model | Historical | SSP1-2.6 | SSP2-4.5 | SSP3-7.0 | SSP5-8.5 |
|---|---|---|---|---|---|
| ACCESS-CM2 | $1^a$ | $1^a$ | $1^a$ | $1^a$ | $1^a$ |
| ACCESS-ESM1-5 | $1^a$ | $3^a$ | $3^a$ | $3^a$ | $1^a$ |
| AWI-CM-1-1-MR | 5 | 1 | | 3 | 1 |
| BCC-CSM2-MR | 3 | 1 | 1 | 1 | 1 |
| CanESM5 | $35^a$ | $35^a$ | $20^a$ | $20^a$ | $20^a$ |
| CESM2 | $11^a$ | $2^a$ | $1^a$ | $3^a$ | $2^a$ |
| CESM2-WACCM | $3^a_p$ | $1^a$ | $1^a$ | $1^a_p$ | $1^a$ |
| FIO-ESM-2-0 | 1 | 3 | | | |
| HadGEM3-GC31-LL | $4^a_p$ | $1^a_p$ | $1^a_p$ | | $3^a_p$ |
| MIROC6 | $10^a_p$ | $1^a_p$ | $3^a_p$ | $3^a_p$ | $3^a_p$ |
| MPI-ESM1-2-LR | 1 | 3 | | 3 | |
| MPI-ESM1-2-HR | $1^a$ | $2^a$ | $1^a$ | $3^a$ | $1^a$ |
| MRI-ESM2-0 | $5^a$ | $1^a_p$ | $1^a$ | $3^a$ | $1^a$ |
| NESM3 | 1 | 2 | 2 | | 2 |
| NorESM2-LM | $1^a_p$ | $1^a_p$ | $1^a_p$ | $1^a_p$ | $1^a_p$ |
| NorESM2-MM | 1 | 1 | | 1 | |
| UKESM1-0-LL | $19^a_p$ | $5^a_p$ | $5^a_p$ | $13^a_p$ | $5^a_p$ |





255  **Appendix A: Appendix: Supplementary plots**



**Figure A1.** (a) A reproduction of Figure 4a and (b, c, d) HWI components illustrated in the same way.







**Figure A2.** (a) A reproduction of Figure 4b and (b, c, d) WCI components illustrated in the same way.



**(a) SSP1-2.6: △T: 2025∼2034**

**(b) SSP1-2.6: △T: 2090∼2099**

**(c) SSP3-7.0: △T: 2025∼2034**

**(d) SSP3-7.0: △T: 2090∼2099**

**(e) SSP1-2.6: △U@500: 2025∼2034**

**(f) SSP1-2.6: △U@500: 2090∼2099**

**(g) SSP3-7.0: △U@500: 2025∼2034**

**(h) SSP3-7.0: △U@500: 2090∼2099**

**(i) SSP1-2.6: △V@850: 2025∼2034**

**(j) SSP1-2.6: △V@850: 2090∼2099**

**(k) SSP3-7.0: △V@850: 2025∼2034**

**(l) SSP3-7.0: △V@850: 2090∼2099**

**(m) SSP1-2.6: △SLP: 2025∼2034**

**(n) SSP1-2.6: △SLP: 2090∼2099**

**(o) SSP3-7.0: △SLP: 2025∼2034**

**(p) SSP3-7.0: △SLP: 2090∼2099**

**Figure A3.** Differences in T, U, V and SLP between future scenarios and present-day. Scenarios shown are SSPs1-2.6 and 3-7.0. Periods chosen are near future (2025-2034) and the end of the 21st century (2090-2099). Difference that is above 90% confidence level is stippled. Boxes indicate areas from which averaged values are used to compute HWI and WCI.