# Peer review of "Competing effects of aerosol reductions and circulation changes for future improvements in Beijing haze"

_Atmospheric Chemistry and Physics, 2021_

## Author Response (AR1)

Review of "Aerosol reductions outweigh circulation changes for future improvements in Beijing haze" by Guo et al.

The authors quantified the role of aerosol emissions and climate in future Beijing haze changes under the SSP scenarios using the latest multi-model simulations (CMIP6) and air pollution indices. They find that haze weather patterns are projected to increase under all the SSPs, which is driven mainly by GHG emission increase. However, local aerosol emissions reduction would be the dominant contributor if PM2.5-related metrics are applied. They highlight the important role of aerosol reductions in future pollution control. This study fits the scope of ACP well and provides some interesting results. Overall, the methods are reasonable, and conclusions are supported by clear illustrations. However, I have the following concerns and hope the authors can resolve them before publication.

1. The title is sort of confusing. Do circulation changes have to improve Beijing haze in the future?

The title has been modified to:

"Competing effects of aerosol reductions and circulation changes for future improvements in Beijing haze".

2. The scientific aim of this study should be further clarified. The authors highlight aerosol reductions outweigh circulation changes for future Beijing haze changes. However, this is not surprising at all. It is commonly known that aerosol emission changes dominate the long-term haze changes, and meteorological indices can't be used to project the real change in air pollution (L175-177).

We agree that our finding that the decrease in aerosol emissions under strong air pollution mitigation scenarios outweighs the continued increase in haze weather patterns and weakening of the winter monsoon for long-term haze changes feels intuitive, and thus not particularly surprising. However, recent observations have run counter to this expectation, and demonstrated a significant role for other drivers in long-term haze changes. Based on current studies, the relationship between aerosol

emissions and the changes in Beijing haze remains unclear (Zhang, Yin et al. 2020, Le, Wang et al. 2020). With reduction in the emissions of anthropogenic aerosol, heavy pollution days over northern China have continued to increase since 2010. With a 90% reduction of key anthropogenic emissions during the 2019 city-lockdown, extreme pollution events simultaneously occurred over northern China. These studies demonstrate that the connection between aerosol emissions and Beijing haze is less clear, with other factors such as meteorology and chemical processes also playing an important role. In light of this, and the fact that aerosol emission reductions also cause changes in local weather patterns (Zhang et al., 2021), we think it is worth exploring near-future changes and their drivers.

In this study, we are focusing on the role of changes in weather patterns. Metrics for weather patterns that can be linked to haze events are valuable tools for examining future changes in haze as they are well-captured by the standard-resolution GCMs that are typically used in climate projections, and have been shown to be well correlated with observed haze events (e.g. Cai, Li et al. 2017, Zhang, Wilcox et al. 2020) .

Figure 4 of the manuscript shows that, with continuing increase of aerosol emissions in SSP3-7.0, the corresponding increase in haze inducive weather pattern is the least among all future scenarios in the near future (2025-2044). This is perhaps counterintuitive, as the increasing aerosol emissions suppresses the increase in the frequency and intensity of Beijing haze in the near future. The aim of our study is to determine whether such meteorological changes are large enough to outweigh the direct effects of near-future emission changes on Beijing haze.

To make the discussions clearer in the manuscript, the following discussion is added in Section 5 (Lines 183-190):

"The opposing impacts of aerosols on these patterns can be seen in the near future (2025-2034) in SSP 3-7.0 where aerosol emissions continue to increase, moderating increases in the haze indices relative to scenarios with aerosol reductions. This

finding is counterintuitive as the increasing aerosol emissions suppresses the increase in the frequency and intensity of Beijing haze in the near future. This finding echo previous studies showing that changing aerosol emissions can feedback onto the meteorological conditions so further complicating the interactions between the two factors."

We have also made changes to the introduction so that it better describes the aims of our study.

3. The authors didn't say anything about nitrate and ammonia aerosols in this study. These two aerosols are increasingly important with the control of SO2 and primary PM2.5. I think some of the CMIP6 models include these two species. A discussion on this issue should be added, at least.

We agree that these species are likely to be playing increasingly important roles in the real world, but unfortunately nitrate aerosol is not simulated in the majority of CMIP6 models. We have added some discussion of the potential role of these species to the manuscript, as follows (Lines 61-65):

"Both NOx and NH3 follow similar pathways to SO2 in future scenarios (Figure S1). However, a decrease in sulphate precursors can benefit nitrate formation, and we would expect an increase in nitrate burden in future (Bellouin, Rae et al. 2011). A simulation that allows nitrate formation would see slower decreases in AOD and in the magnitude of aerosol forcing, so it would act to moderate both the aerosol effects that we discuss in the paper. However, only a few models included the simulation of ammonium nitrate, and discrepancies in the simulation of nitrate remain large (Bian, Chin et al. 2017, Myhre, Samset et al. 2013, and Figure S13 in Turnock, Allen et al. 2020). As a result, the impact of ammonium nitrate is difficult to detect in the CMIP6 ensemble."

4. L15-16: Any reference for the emission changes?

References have been added.

5. L24: "emission reductions of up to 90%" refers to all aerosol emissions? I think only transportation sector declined so much.

Correction has been made between Lines 23-25:

"In 2020, severe haze was formed during the COVID-19 lockdown despite emission reductions of 40% with reduction of up to 90% from the transportation sector (Le et al., 2020)."

6. It is confusing about the title of Section 2.3: Data. You also introduced data information in Section 2.1.

Corrected to "Meteorological and aerosol data"

7. L168: what does "these patterns" mean?

Correction (Lines 183-185):

"The opposing impacts of aerosols on the circulation patterns can be seen in the near future (2025-2034) in SSP 3-7.0 where aerosol emissions continue to increase, moderating increases in the haze indices."

8. Fig.1: Why only SO2 and BC emissions?

Emissions of NOx and NH3 are included in the supplement.

[Figure]

Figure R1. Time series of total emissions for NOx and NH₃ in SSPs1-2.6, 2-4.5, 3-7.0 and 5-8.5 from East Asia, South Asia and global mean.

**Review of** "Aerosol reductions outweigh circulation changes for future improvements in Beijing haze (MS# ACP-2021-198)" by Liang Guo et al.

Based on the calculated PM2.5 concentration and two indices measuring the likelihood of haze in Beijing, this study evaluates the relative role of atmospheric circulation and aerosol emission in determining the future haze in Beijing in CMIP6 models. It suggests that the intensity of aerosol emission overweigh the changes in atmospheric circulation and dominates the future changes in haze days. The results are reasonable. I recommend the authors clarify the following two aspects before I give my next round of recommendations. Details are listed below.

1. At least for me, it is entirely within the expectation that the aerosol emission dominates the haze days when the emission reduces to a certain level. I do not think this conclusion alone is publishable. Nevertheless, it is meaningful to evaluate and explain when the effects of aerosol emission are comparable to those of circulation change in determining the haze days in Beijing.

We agree that our finding that the decrease in aerosol emissions under strong air pollution mitigation scenarios outweighs the continued increase in haze weather patterns and weakening of the winter monsoon for long-term haze changes feels intuitive, and thus not particularly surprising. However, recent observations have run counter to this expectation, and demonstrated a significant role for other drivers in long-term haze changes. Despite reductions in the emissions of anthropogenic aerosol, heavy pollution days over northern China have continued to increase since 2010 (Zhang, Yin et al. 2020). With a 90% reduction of key anthropogenic emissions during the 2019 city-lockdown, extreme pollution events simultaneously occurred over northern China (Le, Wang et al. 2020). These studies demonstrate that the connection between aerosol emissions and Beijing haze is less clear, with other factors such as meteorology and chemical processes also playing an important role.

2. Can the concentration of PM2.5 represent the haze? I think the answer is no. If my understanding is correct, the title and related expressions need changes in the manuscript. If there is no better way to represent haze in CMIP6 models, I suggest the authors adding some discussions to clarify the limitations of this approach.

Thank you for mentioning the difference between the $PM_{2.5}$ and the composition of haze. While PM2.5 does not account for all of the constituents of haze, long-term changes in haze are consistent with long-term changes in PM2.5 (Schichtel et al., 2001). These smaller particles also have a more significant impact on human health than PM10 (e.g. Samek, 2016), and target concentrations are specified for them in both the World Health Organization (WHO) air quality guidelines (WHO, 2005) and Chinese ambient air quality standards (GB 3095-2012). PM2.5 is one of six pollutants monitored by China's Ministry of Environmental Protection, and is used to calculate the Air Quality Index. As PM2.5 provides a clear link between haze and human health, we consider it a benefit, rather than a limitation, to include it in our analysis.

To make the motivation for a specific discussion of PM2.5 clear and confirm the link between PM2.5 and haze, the following has been added to the discussion section of the manuscript (Lines 107-113):

"The particle size distribution of haze varies within a wide range of particle diameters and the $PM_{2.5}$ fraction accounts for a part of this distribution (Wu 2011). While $PM_{2.5}$ does not encompass all the constituents of haze, it is the major factor for impacts on human health and reductions in visibility (An, Huang et al. 2019). As such, it is a key metric in the WHO air quality guidelines (WHO, 2005), and has been adopted as a major index for the air quality standard in many countries (e.g. GB 3-95-2012 in China, 2008/50/EC in Europe). Previous studies have shown that the long-term change in haziness is consistent with changes in $PM_{2.5}$ concentration (Schichtel, Husar et al. 2001). Besides, since haze is not a standard output from CMIP6 models, $PM_{2.5}$ is the best measurement of air quality impact that we have from all of the models."

The title has been modified to:

"Competing effects of aerosol reductions and circulation changes for future improvements in Beijing haze".

Reference:

An, Z., R.-J. Huang, R. Zhang, X. Tie, G. Li, J. Cao, W. Zhou, Z. Shi, Y. Han, Z. Gu and Y. Ji (2019). "Severe haze in northern China: A synergy of anthropogenic emissions and atmospheric processes." Proceedings of the National Academy of Sciences **116**(18): 8657-8666.

Bellouin, N., J. Rae, A. Jones, C. Johnson, J. Haywood and O. Boucher (2011). "Aerosol forcing in the Climate Model Intercomparison Project (CMIP5) simulations by HadGEM2-ES and the role of ammonium nitrate." Journal of Geophysical Research **116**(D20).

Bian, H., M. Chin, D. A. Hauglustaine, M. Schulz, G. Myhre, S. E. Bauer, M. T. Lund, V. A. Karydis, T. L. Kucsera, X. Pan, A. Pozzer, R. B. Skeie, S. D. Steenrod, K. Sudo, K. Tsigaridis, A. P. Tsimpidi and S. G. Tsyro (2017). "Investigation of global particulate nitrate from the AeroCom phase III experiment." Atmospheric Chemistry and Physics **17**(21): 12911-12940.

Cai, W., K. Li, H. Liao, H. Wang and L. Wu (2017). "Weather conditions conducive to Beijing severe haze more frequent under climate change." Nature Climate Change **7**(4): 257-262.

Le, T., Y. Wang, L. Liu, J. Yang, Y. L. Yung, G. Li and J. H. Seinfeld (2020). "Unexpected air pollution with marked emission reductions during the COVID-19 outbreak in China." Science: eabb7431.

Myhre, G., B. H. Samset, M. Schulz, Y. Balkanski, S. Bauer, T. K. Berntsen, H. Bian, N. Bellouin, M. Chin, T. Diehl, R. C. Easter, J. Feichter, S. J. Ghan, D. Hauglustaine, T. Iversen, S. Kinne, A. Kirkevåg, J. F. Lamarque, G. Lin, X. Liu, M. T. Lund, G. Luo, X. Ma, T. Van Noije, J. E. Penner, P. J. Rasch, A. Ruiz, Ø. Seland, R. B. Skeie, P. Stier, T. Takemura, K. Tsigaridis, P. Wang, Z. Wang, L. Xu, H. Yu, F. Yu, J. H. Yoon, K. Zhang, H. Zhang and C. Zhou (2013). "Radiative forcing of the direct aerosol effect from AeroCom Phase II simulations." Atmospheric Chemistry and Physics **13**(4): 1853-1877.

Samek L. (2016). "Overall human mortality and morbidity due to exposure to air pollution. " Int J Occup Med Environ Health 29(3):417-26. doi: 10.13075/ijomeh.1896.00560.

Schichtel, B. A., R. B. Husar, S. R. Falke and W. E. Wilson (2001). "Haze trends over the United States, 1980–1995." Atmospheric Environment **35**(30): 5205-5210.

Turnock, S. T., R. J. Allen, M. Andrews, S. E. Bauer, M. Deushi, L. Emmons, P. Good, L. Horowitz, J. G. John, M. Michou, P. Nabat, V. Naik, D. Neubauer, F. M. O'Connor, D. Olivié, N. Oshima, M. Schulz, A. Sellar, S. Shim, T. Takemura, S. Tilmes, K. Tsigaridis, T. Wu and J. Zhang (2020). "Historical and future changes in air pollutants from CMIP6 models." Atmospheric Chemistry and Physics **20**(23): 14547-14579.

World Health Organization (WHO), (2005). "Air Quality Guidelines Global Update 2005. Particulate matter, ozone, nitrogen dioxide and sulfur dioxide". ISBN 92 890 2192 6

Wu, D. (2011). "Formation and Evolution of Haze Weather." Environmental Science & Technology **34**(3): 157-161.

Zhang, L., L. J. Wilcox, N. J. Dunstone, D. J. Paynter, S. Hu, M. Bollasina, D. Li, J. K. P. Shonk and L. Zou (2020). Future changes in Beijing haze events under different anthropogenic aerosol emission scenarios, Copernicus GmbH.

Zhang, Y., Z. Yin and H. Wang (2020). "Roles of climate variability on the rapid increases of early winter haze pollution in North China after 2010." Atmospheric Chemistry and Physics **20**(20): 12211-12221.